# SealID: Saimaa Ringed Seal Re-Identification Dataset

**DOI:** 10.3390/s22197602

**Published:** 2022-10-07

**Authors:** Ekaterina Nepovinnykh, Tuomas Eerola, Vincent Biard, Piia Mutka, Marja Niemi, Mervi Kunnasranta, Heikki Kälviäinen

**Affiliations:** 1Computer Vision and Pattern Recognition Laboratory (CVPRL), Department of Computational Engineering, Lappeenranta–Lahti University of Technology, 53850 Lappeenranta, Finland; 2Department of Environmental and Biological Sciences, University of Eastern Finland, 80101 Joensuu, Finland

**Keywords:** computer vision, image processing, animal biometrics, re-identification, ringed seals, re-identification dataset

## Abstract

Wildlife camera traps and crowd-sourced image material provide novel possibilities to monitor endangered animal species. The massive data volumes call for automatic methods to solve various tasks related to population monitoring, such as the re-identification of individual animals. The Saimaa ringed seal (*Pusa hispida saimensis*) is an endangered subspecies only found in Lake Saimaa, Finland, and is one of the few existing freshwater seal species. Ringed seals have permanent pelage patterns that are unique to each individual and that can be used for the identification of individuals. A large variation in poses, further exacerbated by the deformable nature of seals, together with varying appearance and low contrast between the ring pattern and the rest of the pelage makes the Saimaa ringed seal re-identification task very challenging, providing a good benchmark by which to evaluate state-of-the-art re-identification methods. Therefore, we make our Saimaa ringed seal image (SealID) dataset (*N* = 57) publicly available for research purposes. In this paper, the dataset is described, the evaluation protocol for re-identification methods is proposed, and the results for two baseline methods—HotSpotter and NORPPA—are provided. The SealID dataset has been made publicly available.

## 1. Introduction

Traditional tools for monitoring animals, such as tagging, require physical contact with the animal, which causes stress and may change the behavior of the animal. Wildlife photo-identification (Photo-ID) provides tools with which to study various aspects of animal populations, such as migration, survival, dispersal, site fidelity, reproduction, health, population size, or density. The analysis task that has gained the most attention is the re-identification of individuals, as it allows one, for example, to study animal migration or to estimate the population size. The basic idea is to collect image data of a species/population of interest by using, for example, digital cameras, game cameras, or crowdsourcing, to identify the individuals and to combine the identification with metadata such as date, time, and GPS location of each image. This enables one to collect a vast amount of data on populations without using invasive techniques, such as tagging. However, the massive image volumes that these methods produce are overwhelming for researchers to go through manually. The large scope of the image data calls for automatic solutions, motivating the use of computer vision techniques. From the image-analysis point of view, the task to be solved is individual re-identification, i.e., finding the matching entry from the database of earlier identified individuals. Although human re-identification has been an active research topic for decades, automatic animal re-identification has recently obtained popularity among computer vision researchers.

The Saimaa ringed seal is an endangered species with around 400 individuals alive at the moment [1]. Due to its conservation status and small population size, automatic computer vision-based monitoring approaches are essential in the development of an effective conservation strategy. In the past decade, Photo-ID has been launched as a non-invasive monitoring method for studying population biology and behavior patterns of the Saimaa ringed seal [2,3]. Ringed seals have a dark pelage ornamented by light grey rings. These fur patterns are permanent and unique to each individual and make re-identification possible.

Saimaa ringed seal image data provides a challenging identification task for developing general-purpose animal re-identification methods that utilize fur, feather, or skin patterns of animals. Large variations in illumination and seal poses, the limited size of identifiable regions, the low contrast between the ring pattern and the rest of the pelage, substantial differences between wet and dry fur, and low image quality all contribute to the difficulty of the re-identification task.

We have compiled an extensive dataset of 57 individuals seals, containing a total of 2080 images with individuals identified in each image by an expert, and made it publicly available at https://doi.org/10.23729/0f4a3296-3b10-40c8-9ad3-0cf00a5a4a53 (accessed on 27 September 2022). See Figure 1 for sample images. In this paper, we describe the dataset, propose the evaluation criteria, and present the results for two baseline methods.

## 2. Related Work

### 2.1. Animal Re-Identification

Camera-based methods utilizing computer vision algorithms have been developed for animal re-identification. Many of them are species-specific, which limits their usability [4,5,6]. There have also been research efforts toward creating a unified approach for the identification of several animal species. For example, WildMe [7,8] is a large-scale project for studying, monitoring, and identifying varied species with distinguishable marks on the body. WildMe’s re-identification methods are based on the HotSpotter algorithm [9]. HotSpotter uses RootSIFT [10] descriptors of affine-invariant regions, spatial reranking with RANSAC, and a scoring mechanism that allows the efficient many-to-many matching of images. This algorithm is not species-specific and has been applied to Grevy’s and plain zebras (*Equus grevyi*), giraffes (*Giraffa*), leopards (*Panthera pardus*), and lionfish (*Pterois*).

Due to the recent progress in deep learning, convolutional neural networks (CNNs) have become a popular tool for animal biometrics [11,12]. For example, re-identification of cattle by using CNNs combined with the k-nearest neighbor classifier was proposed in [13], in which the method was shown to outperform competing methods. The approach is, however, specific to the muzzle patterns of cattle. The muzzle patterns are obtained manually, providing consistent data that simplifies the re-identification. Convolutional neural network (CNN) approaches for animal re-identification by using natural body markings have been applied to various animals including manta rays [14], Amur tigers (*Panthera tigris altaica*) [15,16,17], zebras (*Equus grevyi*), and giraffes (*Giraffa*) [18]. Some species, such as bottlenose dolphins (*Tursiops truncatus*) or African savanna elephants (*Loxodonta africana*) can be identified based on the shape of their body parts, usually their tail or fins, or an ear in case of an elephant. A number of deep learning methods for re-identification are based on this approach: [19], CurvRank [20], finFindR [21], and OC/WDTW [22].

A typical problem in wildlife animal re-identification is that it is practically impossible to collect a large dataset with a large number of images for all individuals. Often, the method needs to be able to identify an individual with only one or a few previously collected examples. Moreover, the animal re-identification method should be able to recognize if the query image contains an individual that is not in the database of the known individuals. Recently, Siamese neural network-based approaches have gained popularity in animal re-identification [23]. These methods provide a tool with which to classify objects based on only one example image (one-shot learning) and to recognize if it belongs to a class that the network has never seen. For example, in [11], the effectiveness of Siamese neural networks for the re-identification of humans, chimpanzees, humpback whales, fruit flies, and octopi was demonstrated.

### 2.2. Saimaa Ringed Seal Re-Identification

A number of studies on the re-identification of ringed seals has been done [24,25,26,27,28,29,30]. In [24], a superpixel-based segmentation method and a simple texture feature-based ringed seal identification method were presented.

In [25], additional preprocessing steps were proposed, and two existing species’ independent individual identification methods were evaluated. However, the identification performance of neither of the methods is good enough for most practical applications. The TOP-1 score is less than 50%, and the TOP-20 score is only 66%. In real life, the re-identification accuracy will be lower because users can upload photos of individuals missing from the database. This means that the manual workload for biologists is still very high, requiring manual analysis of large volumes of images.

In [26], the re-identification of the Saimaa ringed seals was formulated as a classification problem and was solved by using transfer learning. Although the performance was high on the used test set, the method is only able to reliably perform the re-identification if there is a large set of training examples for each individual. Furthermore, the whole system needs to be retrained if a new seal individual is introduced. Finally, it is unclear if the high accuracy was due to the method’s ability to learn the necessary features from the fur pattern, or if it also learned features such as pose, size, or illumination that separated individuals in the used dataset but do not provide the means to generalize the methods to other datasets.

An algorithm for one-shot re-identification of Saimaa ringed seals was proposed in [27]. The algorithm consists of the following steps: segmentation, pattern extraction, and patch-based identification. The first step is done by using end-to-end semantic segmentation with deep learning. The pattern-extraction step relies on the Sato tubeness filter to separate the pattern from the rest of the seal image. The final step is the re-identification. It is done by dividing the pattern into patches and calculating the similarity between them. The patches are compared by using a Siamese triplet network. Overall, the system can identify individuals never seen before and shows promising TOP-5 accuracy, meaning that at least one of the five best matches from the database is correct. This algorithm was presented as a part of a larger, species-agnostic re-identification framework. In [28], a novel pooling layer is proposed to increase the accuracy of patch matching. The idea is to use the eigen decomposition of covariance matrices of features. This method improved the patch and seal re-identification as compared to the previous network architecture in [27].

### 2.3. Re-Identification Datasets

Several publicly available datasets with annotations for animal individuals exist. In [16], a novel large-scale Amur tiger re-identification dataset (ATRW) was presented. It contains over 8000 video clips from 92 individuals with bounding boxes, pose key points, and tiger identity annotations. The performance of baseline re-identification algorithms indicates that the dataset is challenging for the re-identification task.

The ELPephants re-identification dataset [31] contains 276 elephant individuals following a long-tailed distribution. It clearly demonstrates challenges for elephant re-identification such as fine-grained differences between the individuals, aging effects on the animals, and significant differences in skin color.

In [32], the iWildCam species identification dataset was described. The dataset consists of nearly 200,000 images collected from various locations and animal species annotated. However, it should be noted that animal individuals are not identified.

In [33], a manta ray dataset, along with a method for the re-identification of manta rays, was proposed. The training set consists of 110 individuals with 1422 images in total. The test set consists of 18 individuals with 321 images in total. The dataset is challenging for a number of reasons, including large variations in illumination and oblique angles. Those difficulties are similar to the ones encountered in Saimaa ringed seal images.

To form large and varied datasets, crowdsourcing methods can be used. For example, in [18] the authors proposed to use volunteer citizen scientists to collect photos taken in large geographic areas and use computer vision algorithms to semi-automatically identify and count individual animals. The proposed Great Zebra and Giraffe Count and ID dataset contains 4948 images of only two species, the Great Zebra (*Equus quagga*) and the Masai Giraffe (*Giraffa tippelskirchi*). The study in [34] demonstrated the opportunity to collect scientifically useful data from the community through the publicly available photo-sharing platform Flickr by creating a dataset for the Weddell seal (*Leptonychotes weddellii*) species.

## 3. Data

### 3.1. Data Collection and Manual Identification

Data collection was carried out in Lake Saimaa, Finland (61°05′–62°36′ N, 27°15′–30°00′ E) under permits by the local environmental authorities (ELY-centre, Metsähallitus). The Photo-ID data were collected annually during the Saimaa ringed seal molting season (mid-April–mid-June) from the year 2010 to 2019 by both ordinary digital cameras (boat surveys) and game camera traps. The boat surveys were operated in the main breeding habitat of the Saimaa ringed seals during the first years (Haukivesi since 2010 and Pihlajavesi since 2013) and further covered the whole lake since 2016. Powerboats (a 6–8 m powered boat with a 20–60 hp outboard engine, with one to two observers) were used. A minimum distance of 150 m with the observed seal and the used DSLR cameras (a 55–300-mm telephoto lens) for photographing was kept whenever possible. The GPS coordinates, the observation times, and the numbers of the seals were noted. Camera traps were additionally used (Scout Guard SG550 (Bowhunting, Huntley, IL, USA), Scout Guard SG560 (Bowhunting, Huntley, IL, USA), and Uovision UV785 (Uovision Europe, Kangasniemi, Finland)) in Haukivesi (years 2010 to 2012) and Pihlajavesi (since 2013). The game cameras were set in motion sensitivity (2 pictures over a 0.5–2 min time span) or time-lapse (2 pictures every 10 min) and were installed in haul-out locations previously found during the boat survey. In the case of motion-sensitivity cameras, memory cards (2–16 GB) were changed 1 to 3 times a week [2,3]. Seal images were matched by an expert by using individually characteristic fur patterns.

### 3.2. Data Composition

The pelage pattern of the Saimaa ringed seal covers the whole surface area of a seal, making it impossible to see the full pattern from one image. On the other hand, it would be preferable to make a minimal amount of image-to-image comparisons for the re-identification. Ideally, a minimum number of high-quality images to cover the full view of a seal body is wanted as a set of example images for each known individual.

The dataset is divided into two subsets: the database set and the query set. The database is constructed from the aforementioned minimal sets of high-quality images for each individual seal. Images not included in the database set (*N* = 430) are collected into the query set 170 (*N* = 1650). The query images contain the same individuals as in the database. It was also ensured that query images contain some part of a pattern that could be matched to the visible patterns in the database. The database provides a basis for the identification and can be considered as the training set. The query set constitutes the test set used to evaluate the performance of the re-identification methods. The re-identification test set is not seen by the model during training. Typically, the re-identification algorithm searches for the best match from the database for the given query image. The dataset was compiled to be as close to real-life practice as possible.

The total number of individuals, the total number of images in the database and in the query sets, and the minimum, maximum, mean, and median number of images per individual for both sets are presented in Table 1. Image distributions for the database and query sets are illustrated in Figure 2. Example images are shown in Figure 1.

A separate training dataset with matching patches of the pelage pattern is included to provide the basis for training the pattern-matching models. The patch dataset contains, in total, 4599 patches of 60×60 pixels and is divided into the training and test subsets. The training subset contains 3016 images and 16 classes. The test subset contains 1583 images and 26 classes that are different from the classes in the training set. Each class corresponds to one manually selected location on the pelage pattern and each sample from one class was extracted from different images of the same seal. The Sato tubeness filter-based method [27] is applied to each patch to segment the pelage pattern. The extracted pelage pattern patches are manually corrected and included in the dataset. The test set is also divided into the database and query subset with a ratio of 1 to 2. The images that were used to construct the patches dataset are not included in the database and the query subsets of the main re-identification dataset. Examples of patches are presented in Figure 3.

### 3.3. Seal Segmentation

Therefore, the dataset further contains the segmentation masks for each image. The segmentation masks were obtained by using a fine-tuned Mask R-CNN model pre-trained on the MS COCO dataset [35]. The semi-manually segmented datasets of Ladoga and Saimaa ringed seals were used as the ground truth for the transfer learning of the instance seal segmentation [29]. The results of segmentation were further postprocessed in order to fill the holes and smooth the boundaries of segmented seals, and the segmentation masks were manually corrected. An example of a segmented image from the SealID dataset is presented in Figure 4.

## 4. Evaluation Protocol

In this section, an evaluation protocol to enable a fair comparison of methods on the dataset is provided. Although the main task is the re-identification, we provide an evaluation protocol also for the segmentation task to allow the benchmarking of segmentation methods by using the provided segmentation masks.

### 4.1. Segmentation Task

The data for the segmentation task are divided into the training, validation, and test sets. In total, 2080 images are used, and the split is 40% for the training, 20% for the validation, and 40% for the testing. The performance of the segmentation is evaluated by using the intersection over union (IoU) metric defined as
(1)$$\mathrm{IoU}\left(X,Y\right)=\frac{\left|X\cap Y\right|}{\left|X\cup Y\right|},$$
where *X* and *Y* are the sets of pixels from the segmentation result and the ground truth, respectively (see Figure 5).

### 4.2. Re-Identification Task

Top-*k* is the primary metric used for the evaluation of the re-identification task. The re-identification is considered correct if any of the *k* most probable model guesses (best database matches) match to the correct ID. For the model, *f* top-*k* accuracy is defined as
(2)$$\mathrm{top}\text{-}k=\frac{1}{n}\sum_{x_{i}\in X}\left[y_{i}\in f_{k}\left(x_{i}\right)\right],$$
where *X* is the set of samples, $x_{i}$ is the *i*-th sample, *n* is a number of samples, $y_{i}$ is its correct label, $f_{k}\left(\cdot \right)$ is the function that returns *k* most probable guesses for a given sample, and [·] are Iverson brackets, which return 1 if the condition inside is true and 0 otherwise.

Specifically, top-1, top-3, and top-5 metrics are used. The top-1 accuracy is the conventional accuracy, meaning that the most probable answer is correct. The top-*k* accuracy can be viewed as a generalization of this. Typically, a re-identification system is deployed in a semi-automatic manner with a biologist verifying the matches. Providing a small set (e.g., 5) of possible matches speeds up the process considerably, justifying the use of top-3 and top-5 as additional evaluation metrics.

## 5. Baseline Methods

Two baseline re-identification methods were selected. The first one is the HotSpotter [9] algorithm from the WildMe project [8]. It is a unified framework suitable for re-identifying various species with fur, feather, or skin patterns. The second method is NORPPA [30], a Fisher vector-based, pattern-matching algorithm that was developed specifically for ringed seals.

### 5.1. HotSpotter

HotSpotter [9] is a SIFT-based [36] algorithm that uses viewpoint-invariant descriptors and a scoring mechanism that emphasizes the most distinctive keypoints called “hot spots” on an animal pattern. This algorithm has been successfully used for the re-identification of zebras (*Equus quagga*) [9] and giraffes (*Giraffa tippelskirchi*) [18], jaguars (*Panthera onca*) [9], and ocelots (*Leopardus pardalis*) [37]. The method is illustrated in Figure 6.

### 5.2. Seal Re-Identification by Using Fisher Vector (NORPPA)

Novel ringed seal re-identification by pelage pattern aggregation (NORPPA) was proposed in [30]. The pipeline consists of three main steps: image preprocessing including seal segmentation, extraction of local pelage patterns, and re-identification, as shown in Figure 7. The method utilizes feature aggregation inspired by content-based image-retrieval techniques [38]. HesAffNet [39] patches are embedded by using HardNet [40] and aggregated into Fisher vector [41] image descriptors. The final re-identification is performed by calculating cosine distances between Fisher vectors.

## 6. Results

The results for the HotSpotter and the NORPPA algorithms are presented in Table 2. For both approaches, the experiments were executed with and without the preprocessing step, which consists of tone mapping and segmentation as described in [30]. It is clear that preprocessing improves the results for HotSpotter, but it is even more important for NORPPA. Even though the accuracy of NORPPA without the preprocessing step is much lower than that of HotSpotter, with the preprocessing step NORPPA outperforms HotSpotter by a notable margin. Examples of the results of the NORPPA and HotSpotter algorithms are presented in Figure 8 and Figure 9, respectively.

## 7. Conclusions

In this paper, the Saimaa ringed seal re-identification dataset (SealID) was presented. Compared to other published animal re-identification datasets, the SealID dataset provides a more challenging identification task due to the large variation in illumination and seal poses, the limited size of identifiable regions, low contrast of the ring pattern, and substantial variations in the appearance of the pattern. Therefore, the database allows us to push forward the development of general-purpose animal re-identification methods for wildlife conservation. The dataset contains a curated gallery database with example images of each seal individual and a large set of challenging query images to be re-identified. The segmentation masks are provided for both database and gallery images. A separate dataset of pelage pattern patches is included in the database. We further propose the evaluation protocol to allow a fair comparison between methods and show results for two baseline methods—HotSpotter and NORPPA. The results demonstrate the challenging nature of the data, but also show the potential of modern computer vision techniques in the re-identification task. We have made the database publicly available for other researchers.

## Figures and Tables

**Figure 1 sensors-22-07602-f001:**
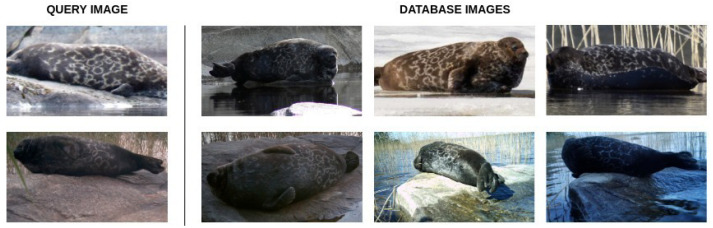
Examples from the database and the query datasets. Each row contains images of an individual seal. For each image from the query dataset (**left**), there exists a corresponding subset of images from the database (**right**).

**Figure 2 sensors-22-07602-f002:**
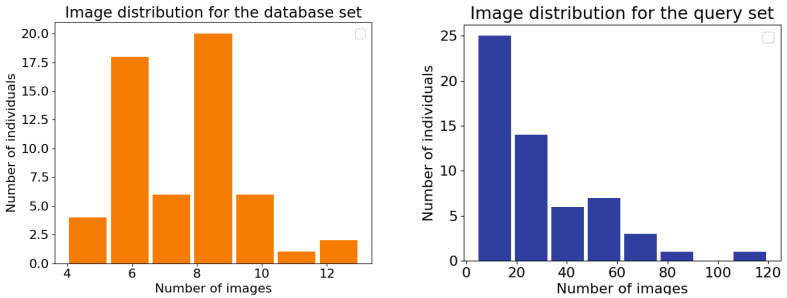
Image distributions for the database (**left**) and query sets (**right**). For example, in the query dataset (**right**), 25 individuals (*y*-axis) had less than 20 images (*x*-axis).

**Figure 3 sensors-22-07602-f003:**
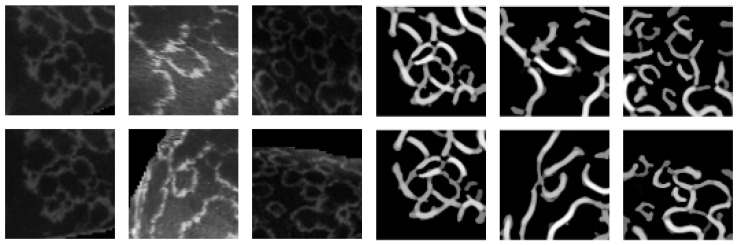
Examples of patches. Original patches (**left**). The corresponding pattern patches (**right**).

**Figure 4 sensors-22-07602-f004:**
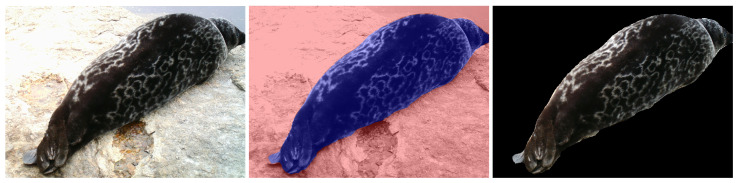
An example of the seal segmentation by using the mask. The original image (**left**), the mask highlighted in blue and the background highlighted in red (**middle**), and the result of the segmentation (**right**).

**Figure 5 sensors-22-07602-f005:**
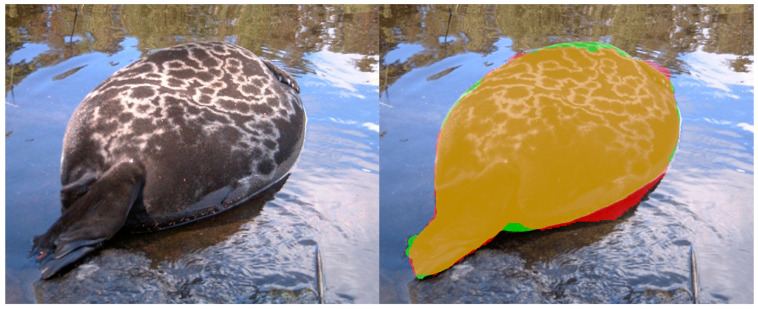
Segmentation mask example. The green color depicts the ground truth, red is the segmentation result, and yellow is the intersection.

**Figure 6 sensors-22-07602-f006:**
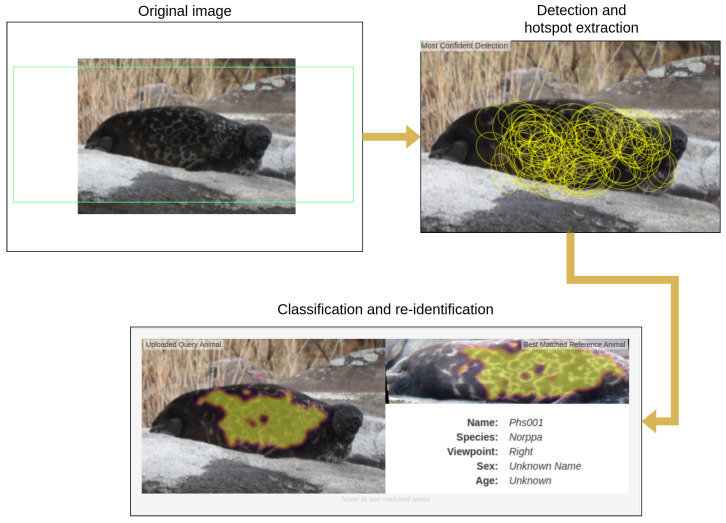
The HotSpotter method applied to an image of a Saimaa ringed seal. Yellow circles represent the extracted features or “hot spots”. The matching areas between two images, are highlighted with yellow. The matching image is chosen according to the score.

**Figure 7 sensors-22-07602-f007:**
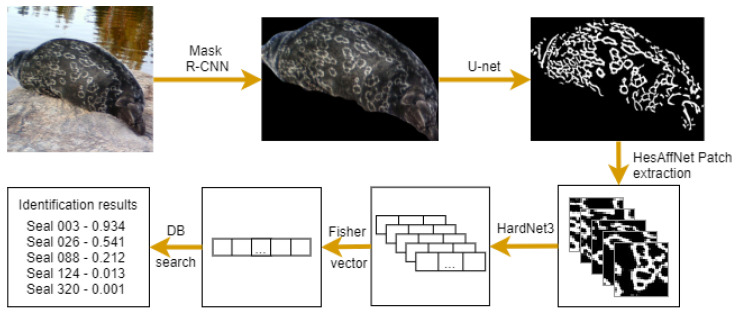
NORPPA re-identification pipeline [30].

**Figure 8 sensors-22-07602-f008:**
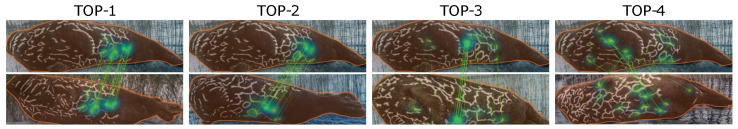
TOP-4 examples for the NORPPA algorithm. First line: query image (phs10), Second line: four best matches in decreasing order of similarity from left to right. Matched hotspots are highlighted in green. TOP-1–TOP-3 matches are correct. TOP-4 is incorrect.

**Figure 9 sensors-22-07602-f009:**
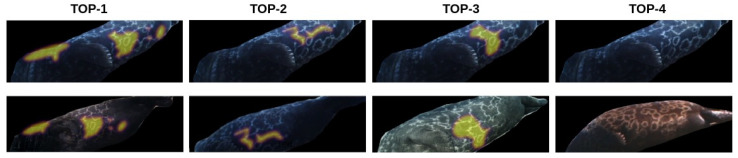
TOP-4 examples for the Hotspotter algorithm. First line: query image (phs10), Second line: four best matches in decreasing order of similarity from left to right. Matched hotspots are highlighted in yellow. TOP-1–TOP-3 matches are correct. TOP-4 is incorrect.

**Table 1 sensors-22-07602-t001:** Image distributions for the database and the query sets.

	Database Set	Query Set
Total number of images	430	1650
Min	4	4
Max	13	120
Mean	7.5	28.9
Median	6	42
Total amount of individuals	57	

**Table 2 sensors-22-07602-t002:** Re-identification test set based on top-k accuracy results with HotSpotter and NORPPA. The best results are presented in bold.

		TOP-1	TOP-3	TOP-5
HotSpotter	Raw	61.87%	63.63%	64.42%
	Preprocessed	69.39%	72.00%	73.15%
NORPPA	Raw	49.52%	59.58%	64.55%
	Preprocessed	**77.64%**	**82.97%**	**85.27%**

## Data Availability

All data and materials are publicly available at https://doi.org/10.23729/0f4a3296-3b10-40c8-9ad3-0cf00a5a4a53 (accessed on 27 September 2022). The codes for the described experiments are available at https://github.com/kwadraterry/Norppa (accessed on 27 September 2022).

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
