# Peer review of "SealID: Saimaa Ringed Seal Re-Identification Dataset"

_sensors, 2022, doi:10.3390/s22197602_

Round 1
Author Response
We would like to thank the reviewer for their valuable comments. We have done our best effort to address them all. Please see the attachment.

Reviewer 2 Report
The link provided in the abstract is not working. It redirects to:
https://etsin.fairdata.fi/dataset/22b5191e-f24b-4457-93d3-95797c900fc0
and the page says: "Sorry! The page couldn't be found." Therefore, I cannot access the dataset.
Some statements should be removed from the abstract and placed into indroductionary section: "However, massive image volumes that these methods produce are overwhelming for researchers to go through manually which calls for automatic systems to perform the analysis. The analysis task that has gained the most attention is the re-identification of individuals, as it allows, for example, to study animal migration or to estimate the population size."
Section 1. (Introduction) should be a little bit improved.
Section 2. (Related work) is state-of-art.
Section 3. (Data) is solid and in my opinion needs no change.
Section 4. (Evaluation protocol) should be a little bit improved. Although it is really nice to see "40% for the training, 20% for the validation, and 40% for the testing" as I mostly receive works with only validation which is not a really valid measure, like test results.
Section 5. (Baseline methods) could be improved by providing some values for architecture on Figure 7 (although it is not obligatory requirement).
Results in section 6 are clearly presented.
Conclusion section should be a little bit improved.
Author Response

(The authors gave the same response as above.)

Reviewer 3 Report
The authors collected a Saimaa ringed seal re-identification dataset. This work has application values; however, the authors just conducted a preprocessing on two existing methods on their dataset. One existing method [30] is proposed the author on ArXiv. Thus the academic value is very low. Similarly, the authors can also collect the datasets for other animal species, each dataset can be introduced in a separate paper, but the methods are not improved. The authors should improve the methods and make sure that the improved methods are more suitable for the dataset.
Round 2
Reviewer 3 Report
(1)Please check the manuscript carefully to remove the typos, improve the language and format. E.g.
“Fig. 7..” (Line 244) One dot is redundant
(2)Mask R-CNN is used in Fig. 7, actually some SOTA similar methods can be introduced or cited in Section 2.1 or 3.3, such as: - Mask Refined R-CNN: A network for refining object details in instance segmentation
